# Local probe of single phonon dynamics in warm ion crystals

A. Abdelrahman[1,2], O. Khosravani[1], M. Gessner[3,4,5], A. Buchleitner[3,6], H.-P. Breuer[3], D. Gorman[1], R. Masuda[1], T. Pruttivarasin[1,7], M. Ramm[1], P. Schindler[1] & H. Häffner[1]

The detailed characterization of non-trivial coherence properties of composite quantum systems of increasing size is an indispensable prerequisite for scalable quantum computation, as well as for understanding non-equilibrium many-body physics. Here, we show how autocorrelation functions in an interacting system of phonons as well as the quantum discord between distinct degrees of freedoms can be extracted from a small controllable part of the system. As a benchmark, we show this in chains of up to 42 trapped ions, by tracing a single phonon excitation through interferometric measurements of only a single ion in the chain. We observe the spreading and partial refocusing of the excitation in the chain, even on a background of thermal excitations. We further show how this local observable reflects the dynamical evolution of quantum discord between the electronic state and the vibrational degrees of freedom of the probe ion.

[1] Department of Physics, University of California, Berkeley, California 94720, USA. [2] Department of Physics, Stanford University, 452 Lomita Mall, Stanford, California 94305, USA. [3] Physikalisches Institut, Albert-Ludwigs-Universität Freiburg, Hermann-Herder-Str. 3, D-79104 Freiburg, Germany. [4] QSTAR, INO-CNR and LENS, Largo Enrico Fermi 2, I-50125 Firenze, Italy. [5] Istituto Nazionale di Ricerca Metrologica, Strada delle Cacce, 91, I-10135 Torino, Italy. [6] Keble College, University of Oxford, OX1 3PG Oxford, UK. [7] Department of Physics, Mahidol University, 272 Rama VI Rd., Ratchathewi, Bangkok 10400, Thailand. Correspondence and requests for materials should be addressed to A.A. (email: mabd@kth.se) or to H.H. (email: hhaeffner@berkeley.edu).

The faithful description of the state of an interacting many-particle quantum system turns ever more difficult as the system size increases[1–3]. Therefore, one usually focuses on collective quantifiers—such as, for example, the total magnetization of a spin chain—to distinguish macroscopically distinct many-body phases. However, engineered many-body quantum systems, such as trapped ions, cold atoms and superconducting circuits, offer the unique and distinctive feature that their microscopic structure is accessible to experimental diagnostics[4–12]. Therefore, hitherto unaccessible dynamical features can be directly observed offering a novel opportunity to gain insight into the emergence of macroscopically robust features[13–18]. Further, the system size can be increased starting from only a few particles up to millions while maintaining coherent dynamics. Thus, those engineered systems allow one to study the emergence of macroscopic properties directly from the microscopic spectral and dynamical structure[6,8].

On the other hand, a microscopic approach where not only global observables but also the local ones as well as their correlations become an essential part of the description will at some point hit a complexity threshold. In particular, the increasing number of interacting degrees of freedom implies a rapidly proliferating number of possible defects, which induce disorder and/or noise; with finite temperature effects as one of the most fundamental and ubiquitous perturbations. Consequently, there is a conceptual and practical need to develop scalable and experimentally feasible approaches[19–22] to probe well-defined microscopic quantum features such as correlation functions and quantum correlations even at system sizes which prevent an exhaustive microscopic characterization, and where the presence of uncontrolled noise such as a finite temperature is unavoidable. Such methods will help to gain a better understanding of the ultimate realm of non-trivial quantum mechanical effects, on meso- and macroscopic scales. We believe that such methods could also be useful to characterize the naturally very complex functioning of quantum computing devices and thus help push quantum computing architectures to scales which become practically relevant.

To contribute to this endeavour, we present a tagging method which can be used to track single quantum excitations in warm quantum many-body systems. We achieve this by coupling an auxiliary qubit in form of the electronic state of a probe ion to the motion of an ion chain of variable length[23–25]. We will demonstrate that this allows one to measure the autocorrelation function of the phonon dynamics even in situations where only access to a subsystem is available in refs 20,21. Moreover, we will see that the local dynamics of the qubit is directly linked to the dynamical evolution of the quantum discord[26] between the qubit degree of freedom of the probe ion and its local vibrational degree of freedom. We recall that discord describes local quantum

properties of correlated states[22,26] which are relevant for quantum information tasks such as entanglement distribution[27] and activation[28,29], as well as for local quantum phase estimation[30].

## Results

**Experimental system.** The dynamics of the transverse motion $(x, y)$ of an ion string aligned along the $z$ axis can be described by the Hamiltonian of a chain of coupled quantum harmonic oscillators[31,32],

$$H_{\text{chain}} = \sum_{i=1}^{N} \hbar \omega_{x,i} a_i^\dagger a_i + \hbar \sum_{i=1}^{N} \sum_{\substack{j=1 \\ j<i}}^{N} t_{ij} \left( a_i^\dagger a_j + a_i a_j^\dagger \right), \quad (1)$$

where $a_i$ is the annihilation operator acting on the Fock state of the $i$th harmonic oscillator which represents the corresponding ion's transverse motional state and the $\omega_{x,i}$ are the frequencies of the transverse motion of each ion along the $k$-vector of the qubit laser, which are modified by the Coulomb potential due to the interaction with neighbouring ions. These frequencies are given by $\omega_{x,i} = \omega_x - \sum_{\substack{j=1 \\ j\neq i}}^{N} t_{ij}$ in which $\omega_x$ is the transverse trap centre-of-mass frequency and $t_{ij}$ is the coupling between the local modes of ions $i$ and $j$,

$$t_{ij} = \frac{1}{2} \frac{1}{m\omega_x} \frac{e^2}{4\pi\epsilon_0} \frac{1}{|z_i^0 - z_j^0|^3} \quad (2)$$

with the equilibrium position $z_i^0$ of the $i$th ion along the $z$ axis.

We trap strings between 8 and 42 $^{40}$Ca$^+$ -ions in a linear Paul trap (see Fig. 1) with trapping frequencies of $\omega_x = 2\pi \times \sim 1.9$ MHz, $\omega_y = 2\pi \times \sim 2.2$ MHz and $\omega_z$ ranging from $2\pi \times 94$ to $2\pi \times 196$ kHz.

The string is cooled using laser light at 397 nm, red detuned with respect to the $S_{1/2} \leftrightarrow P_{1/2}$ transition, to the Doppler limit characterized by a mean phonon number $\bar{n} \approx 5$ of the transverse modes. To control the quantum information in the string, we drive the quadrupole transition between the $|S\rangle = |S_{1/2}, m_j = -1/2\rangle$ and $|D\rangle = |D_{5/2}, m_j = -1/2\rangle$ states by a 729 nm laser beam[33], as schematically shown in Fig. 1. This beam is tightly focused addressing the probe ion at the end of the ion chain, with a small projection on the $z$ axis, and equal projections on the $x$ and $y$ axis. To create correlations between the electronic and motional degree of freedom, a short laser pulse $R^+(\theta, \varphi)$ with length $\theta$ and phase $\varphi$ tuned to the $+\omega_x$ sideband of the quadrupole transition induces the coupling $|S, n\rangle \leftrightarrow |D, n+1\rangle$, where $n$ labels a local motional Fock state of the addressed ion[34,35]. To initially prepare a localized vibrational excitation, the pulse duration of 5–10 μs needs to be much shorter than the inverse coupling $t_{12}$ of the local motion to the adjacent ion.

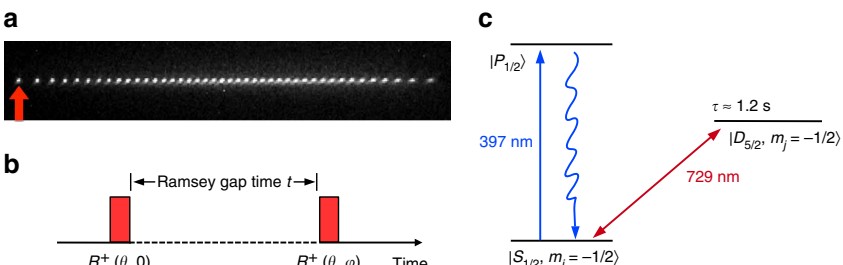

**Figure 1 | Schematic overview and electronic structure of $^{40}$Ca$^+$.** (**a**) Image of 42 ions. The red arrow indicates the position of the laser beam exciting the probe ion on the blue motional sidebands of the transverse motion with light at 729 nm. (**b**) Ramsey sequence with a free evolution time $t$ governed by the Coulomb interaction between the ions. (**c**) relevant electronic levels of $^{40}$Ca$^+$. The quadrupole transition between the $|S_{1/2}, m_j = -1/2\rangle$ and $|D_{5/2}, m_j = -1/2\rangle$ states is driven by a narrow linewidth 729 nm laser beam tightly focused on the probe ion. The Zeeman degeneracy is lifted by applying a magnetic field of 323 μT. Cooling and readout are performed on the $|S_{1/2}\rangle \rightarrow |P_{1/2}\rangle$ transition at 397 nm.

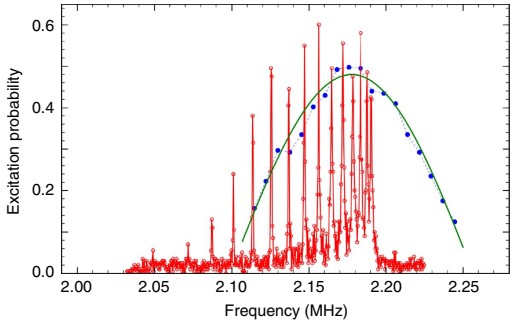

**Figure 2 | Spectra of blue transverse sidebands of a 42-ion string.** Excitation probability to the $|D_{1/2}, m_j = -1/2\rangle$ state of the probe ion as a function of the detuning. Red: excitation time of 900 μs, together with a relatively low intensity resolves individual normal modes. Visible are the normal modes of the x direction. Blue: a short high-intensity pulse of length 8 μs excites superpositions of the normal modes. Green: fit of the excitation to a sinc-function with only the amplitude and centre frequency as free parameters.

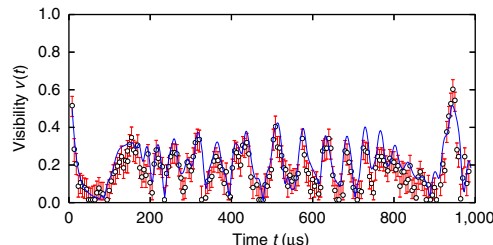

**Figure 3 | Visibility measurement for 42 ions.** Visibility (according to equation (3)) deduced from the population evolution of the state $|D_{1/2}, m_j = -1/2\rangle$ as a function of the free evolution time $t$ of the Ramsey sequence. A partial revival of the initial state population occurs at the rephasing times of the eigenfrequencies. Experimental results (red circles) are shown along with theory (blue line), for a chain of 42 ions with an axial trapping frequency $\omega_z = 2\pi \times 106.9$ kHz. Error bars represent the Bayesian 90% credible interval for the visibility. The only free parameter in the fit is an overall scale factor of the visibility of 0.67 to take into account loss of coherence mainly due to the incoherent background of the laser light (see text).

Figure 2 shows a spectrum near the blue sideband of the x and y modes of a 42-ion string with axial frequency $\omega_z = 2\pi \times 72.14$ kHz. For long excitation times of 900 μs (red trace in Fig. 2), we resolve the normal modes of the 42-ion chain.

However, a short excitation time of 8 μs at about 250 times the intensity used to resolve the normal mode spectrum excites superpositions of the normal modes (blue trace in Fig. 2) corresponding to the desired local excitation of the ion string[32].

After this initial state preparation, we allow the initially localized phonon to travel into the phonon bath of the warm ion string before a second probe pulse interrogates the coherence between the qubit and the vibrational motion (Fig. 1)[36]. Full interference contrast can only be restored if the original vibrational excitation refocuses at the probe ion simultaneously with the probe pulse. The observation of Ramsey fringes as a function of the relative phase between the two pulses can furthermore be traced back to qubit-phonon discord of the probe ion[22,36] (Supplementary Notes 1–3).

**Theoretical interpretation.** For the probe ion initially in the state $|S, n\rangle$, the preparation pulse $R^+(\theta, 0)$ of length $\theta$ and phase 0 creates a superposition of the form $\alpha_n |S, n\rangle + \beta_n |D, n+1\rangle$, where $\alpha_n$ and $\beta_n$ are specific for each Fock state $|n\rangle$. The inverse operation can be applied with a second pulse, with the same parameters but opposite phase $\Delta\varphi = \pi$ with respect to the first pulse.

As long as the system has not evolved between both pulses, we expect full contrast. However, if the phonon created by the preparation pulse is no longer localized at the probe ion, the second pulse (Fig. 1) cannot map the qubit-phonon coherences back to the electronic populations and therefore no phase dependency can be observed. Thus, loss of phase contrast indicates delocalization of the phonon excitation that is tagged by the probe ion's internal state (Supplementary Notes 1–3 for a detailed derivation).

More specifically, the phase contrast or visibility

$$v(t) = \frac{\max_\varphi \left( P_D(t) - \min_\varphi (P_D(t)) \right)}{\max_\varphi \left( P_D(t) + \min_\varphi (P_D(t)) \right)}, \qquad (3)$$

where $P_D$ is the probability that after the Ramsey sequence with free evolution time $t$ the ion is found in the excited $|D_{1/2}, m_j = -1/2\rangle$ state. The visibility $v(t)$ is closely connected

to both the autocorrelation function of the probe ion's vibrational degree of freedom and to the discord between the electronic and the motional degrees of freedom of the probe ion (equations (19), (29) and (32) of Supplementary Notes 1 and 3). The reason for this is that all three quantities relate to the coherences between the electronic states $|S\rangle$ and $|D\rangle$ in the density matrix pertaining to the motional degrees of freedom of the probe ion.

To make a quantitative connection, we assume that the density matrix describing the ion string after laser cooling is diagonal in the collective mode basis and that all modes are equally populated. This is justified by noting that laser cooling acts on the individual ions on timescales faster than the coupling between ions and that the spread of the eigenmode frequencies are nearly degenerate.

With these assumptions, we find in Supplementary Note 2 (equations (19) and (22) therein) for the modulus of the correlation function $|C(t)|$

$$|C(t)| = \left| \left\langle a_1(t) a_1^\dagger(0) \right\rangle \right| = (\bar{n} + 1) v(t), \qquad (4)$$

where $\bar{n} = \left\langle a_i^\dagger a_i \right\rangle \approx 5$ is the average number of phonons in each of the normal modes. Further, we arrive at a simple measure for the quantum discord (equation (32) of Supplementary Note 3):

$$D(t) = \frac{\pi}{4} \frac{|C(t)|}{\bar{n} + 1} = \frac{\pi}{4} v(t). \qquad (5)$$

To derive this relation, we have performed a perturbation expansion for small $\theta/2 \approx 0.3$, including contributions of first and second order, and used the additional assumption that the initial phonon state may be approximated by an equilibrium state of the local phonon modes (Supplementary Note 3.

**Experimental results.** For the experiments, we first determine the time it takes to excite the probe ion to the $|D\rangle$-level with a probability of 0.5, nearly saturating the transition for finite temperature ion strings. We call this time the effective π-time $t_\pi$. The actual sequence to probe the dynamics of the crystal consists of two pulses each of length $t_\pi/2$ (Fig. 1) and separated by time $t$. To detect and quantify the discord between qubit and local vibrational degree of freedom, we vary the phase difference $\Delta\varphi$ between the first and the second Ramsey pulses. In particular, we choose $\Delta\varphi = \{0, \pi/2, \pi, 3\pi/2\}$ and extract the phase contrast $v(t)$ for each value of the free evolution time $t$. Figure 3 shows a comparison of the experimentally inferred phase contrast for

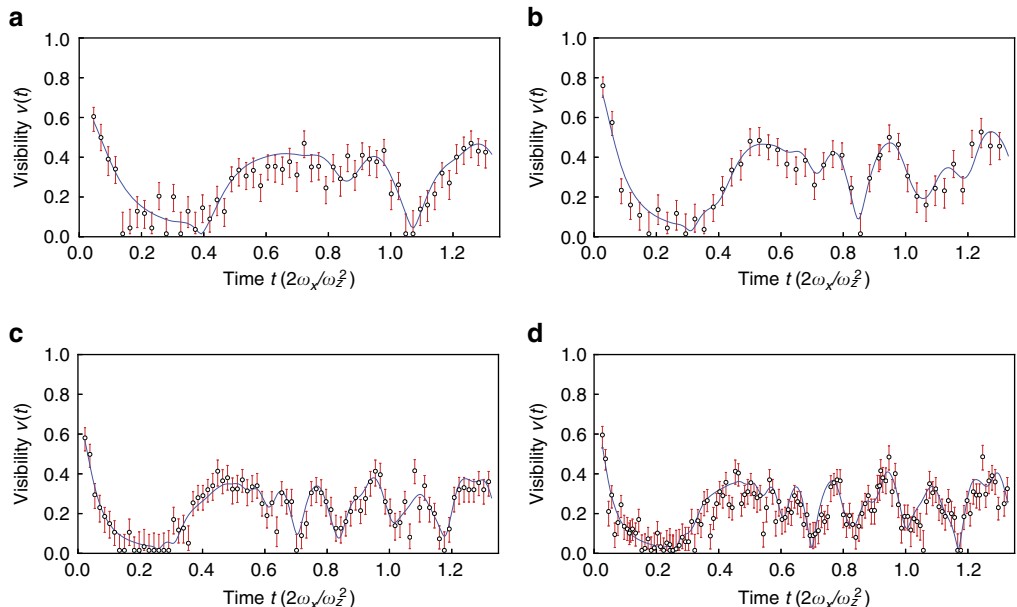

**Figure 4 | Visibility measurements for a variable number of ions.** Experimental result in **a**, is for 8 ions, (**b**), 14 ions, (**c**), 25 ions and **d**, with 33 ions where the axial trapping frequency takes the values $\omega_z = 2\pi \times$ (195.8, 155.2, 93.8 and 117.1 kHz), respectively. Error bars represent the Bayesian 90% credible interval for the visibility. While the traces are very similar, increasing the number of ions tends to produce sharper features.

42 ions to the theoretical expectations as given by equation (19) in Supplementary Note 1.

To gather sufficient statistics, we repeat the experiment 500–750 times per data point. For short times $t$, we observe a large phase contrast as the phonon excitation is still localized at the site of the probe ion. After a few tens of microseconds, the phonon excitation couples to the other sites[32] and the phase contrast diminishes. However, the phase contrast revives when the phonon excitation recurs at the original site. This revival of the phase contrast also proves that the phase coherence of the initially injected phonon is maintained even as it delocalizes over the ion string. We also study this dynamics for 8, 14, 25 and 33 ions (Fig. 4).

Common to all measurements is a rapid loss of phase contrast followed by a specific revival pattern. Comparison with theory shows that the revival pattern is governed by the phonon dynamics in the ion chain. However, we also observe a reduction of the maximally expected visibility from 1 to values between 0.66 and 0.80. The reduction is independent of the Ramsey gap time, consistent with the measured Gaussian qubit decoherence with $T_2 \sim 2$ ms. Instead, we attribute the reduced contrast mainly to a broad incoherent background of the laser light (see Methods section).

## Discussion

We expect that suitable extensions of the scheme to protocols consisting of more than one ion or two pulses are in fact capable of extracting higher-order phonon correlation functions both in space and time. This way, the methods of non-linear spectroscopy, which are typically employed to study dynamical and spectral features of molecular aggregates and semiconductors[37], become accessible to probe quantum optical many-body systems. This opens up a powerful way of analysing complex interacting quantum systems by measuring space and time correlations[20,21].

In this context, it is also interesting to note that the method tracks an excitation in a bath of substantial size. In the example of a 42-ion string cooled down to the Doppler limit of about five

local transverse quanta, there are about $42 \times 5 \approx 200$ phonons present. Our method generates a single phonon and tags it with a specific phase relationship to the electronic state of the probe ion, allowing us to re-identify this excitation when it returns to its origin. Hence, the method allows to follow the dynamics of single phonons in a finite temperature environment.

Our measurements also show that quantum coherence of motional excitations can be preserved even in long ion strings. Direct extensions of this work are to measure how phonons scatter off impurities and how this affects the transport dynamics in a finite temperature environment[38]. This could be done by coupling some of the phonons to individual qubits of other ions via sideband interactions. These interactions implement a non-linear dynamics in the photon bath allowing one to study more complex cases than the linear dynamics studied here. Further extensions include perturbing the potential with an optical lattice enabling studies of nanofriction and the Aubry-transition[39] or the interplay of disorder and interactions[40].

More generally, it would be interesting to study how this method could be applied to other systems, for example, interacting spin chains, where individual excitations could be coherently traced via an auxiliary probe spin.

Finally, our method is scalable in the sense that the control of the subsystem is independent of the size of the total system. Thus, the discussed method can be applied or adapted to large coherent systems where suitable single particle control is available, such as for instance for neutral atoms in optical lattices[5].

## Methods

**Decoherence due to laser light.** We observe a reduction of the maximally expected visibility from 1 to between 0.66 and 0.80. The reduced contrast is mainly due to a broad incoherent background on the laser light on the order of $B = -40$ dB below the carrier of the laser light, however, peaking to around $B = -28$ dB near the bandwidth of 700 kHz of our servo loop to stabilize the laser frequency. This incoherent background drives other transitions than the blue sideband, and in particular the strong atomic carrier transitions between the $S_{1/2}(m_J = -1/2)$ and $D_{5/2}(m_J = \{-5/2, -3/2, -1/2, 1/2 \text{ and } 3/2\})$ levels. Our geometry was not optimized to drive only the $S_{1/2}(m_J = -1/2)$ and $D_{5/2}(m_J = -1/2)$ transition, and hence we assume that all carrier transitions

have a similar coupling strength of $\Omega_0 \approx 150$ kHz when driving effective $\pi$-pulses on the blue sideband in $t_\pi = 20$ μs. Thus, the population of the D-state will increase by $\sin(\sqrt{B}\Omega_0 t_\pi/2)$, corresponding to an increase between 0.016 ($B = -40$ dB) and 0.06 ($B = -28$ dB) for each $\pi_{\text{eff}}/2$-pulse and transition. We estimate that this effect accounts for 0.2 of the loss of contrast for both pulses and all five atomic carrier transitions.

Another concern is motional heating during the free evolution time. In our trap, motional heating almost exclusively stems from voltage noise at the trap electrodes resulting in common noise to all ions. Thus, the centre-of-mass mode heats with about 0.2 quanta per ms while all other modes heat with less than 0.01 quanta per ms. Because the heating is predominantly common mode, motional heating will not influence the intrinsic dynamics of the ion string.

**Data availability.** All relevant data are available from the authors.

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

## Acknowledgements

This work has been supported by AFOSR through the ARO Grant No. FA9550-11-1-0318 and by the NSF CAREER programme grant no. PHY 0955650. A.B. and H.-P.B. acknowledge support through the EU Collaborative Project QuProCS (Grant agreement 641277).

## Author contributions

H.H., M.G., M.R. and T.P. conceived the experiment. A.A., M.R., T.P., R.M., D.G. and P.S. carried out the measurements. O.K., M.G., H.-P.B. and A.B. carried out the theoretical analysis, A.A., H.H., O.K., M.G., H.-P.B. and A.B. wrote the manuscript. All authors contributed to the discussions of the results and manuscript.

## Additional information

**Competing interests:** The authors declare no competing financial interests.

**Publisher's note**: 

