## [Peer Review File · Nature Communications]

Reviewers' Comments:

Reviewer #1 (Remarks to the Author)

The authors present a method and experimental results on which they can measure correlation functions of a chain of coupled harmonic oscillators by coupling on of them to a local 2-dimensional quantum system. They work with linear ion chains of variable length trapped in a radio-frequency Paul trap. The radial motion of each ions couples to the others through the coulomb potential. As a local probe they use the spin degree of freedom of one of the extremal ions which they can control and measure with a strongly focused laser beam.

They can see how an initial excitation propagates along the chain and refocuses on the initial ion at the eigen frequencies of the collective motion.

From a classical perspective the experiment is quite simple minded and easy to understand. Take a long chain of coupled oscillators, kick on of them, extract properties from the whole chain by monitoring only this ion. However the implementation involves the interaction of the electronic state of one of the ions with the local motional modes and generates quantum correlations. As mentioned by the authors, this is an interesting platform in which to study quantum systems coupled to environments, which in the future could be more complex involving defects and coupled environments.

Also, an outstanding feature of this experiment is that they can observe these effects in a "warm" ion crystal where a single quanta of motion is seen propagating along an ion crystal with mean phonon number of 5.

The method seems to be inspired in an extension of 2d spectroscopy techniques applied to ion chains Refs. 21 -23, and a continuation work on what was published by the authors on Ref. 36.

I enjoyed reading the paper. I feel the body of the paper clearly and easily written and tell an interesting coherent story. I found the though, abstract and first two paragraphs, did not flow as easily.

For example, the first sentence. Though the coherence properties of composite quantum systems are of interest to quantum computation, and this paper studies the propagation of a perturbation in a system of oscillators measured by a 2d quantum system, it seems too much to directly relate the importance of this paper to that. On the other hand, the connection to the study of many-body physics is very well justified.

Following are some specific questions and comments I feel should be addressed.

1) The measurement protocol should be explained somewhere, probably as a subsection in the Methods section. I understand they are using the techniques of ion shelving and fluorescence detection, because of the citation of ref. 33, but some explanation should be available for readers not acquainted with this technique.

- A detailed description of what "excitation" in Fig. 2 is, should be provided, specifically indicating if excitation is measured on the "probe" ion or on the whole chain, or however it was done.

- Also I cant find a proper definition of the quantity P_D , which, to me, is clearly Probability of the ion in the D state after the experiment, but should be explicitly written somewhere.

2) The overall quality of the figures and the data presented is a bit messy.

Figure 1 looks like it was prepared in 5 minutes. The fonts sizes are not balanced and the non

clear division into panels makes it a bit uneasy to read.

Figure 3 shows results obtained for 4 different ion chain lengths. I ask myself, what happened in subfigure 3c), why weren't measurements taken for the full time scale. This looks very untidy. I would either recommend the authors to re-measure this case or at the worse to cut the other 3 plots so that the figure doesn't look like "preliminary data".

3) A stylistic comment, which I am not sure about, but the editors will tell.

The sentence "In the supplement, we then find for the modulus of autocorrelation function $|C(t)|$.." doesn't seem to be in the style results from the Supplementary material are referenced.

3b) Continuing on that sentence, but now on the physics.

There is no discussion of the applicability and universality of relations $|C(t)|$ and $D(t)$ in the main text. This is slipped under the carpet with the reference to the Supp. Mat.

I also wonder, specifically how good are the approximations made to obtain these simple expressions. In particular, the assumption of small excitations. Its validity for a $\pi/2$ pulse, which makes the "expansion parameter" of a value 0.3, as mentioned in the supplementary material is not clear to me. I would like to know, for example if this assumption over or under estimates the results.

Also, I wonder if these assumptions are made for the theoretical predictions plotted with the experimental results, or if these are the full theoretical results. And in such a case, what the deviation is.

4) In the main text it is not clear what expectation value of n is. The answer can be digged from the supplementary material, but a small hint could be provided.

5) The authors state that the initial fast reduction of phase contrast is attributed to a broad incoherent background of the laser light (see methods). The explanation provided seems sound. I wonder though, why for most cases the initial visibility is 0.6 and for the 14-ion chain it is significantly higher.

6) Is no heating considered? If so, please state why under the experimental conditions this is reasonable.

Reviewer #2 (Remarks to the Author)

The manuscript by Abdelrahman & co-workers describes the experimental probing of the phonon dynamics in a one-dimensional ion chain. The probing technique is based on a Ramsey scheme developed a few years ago in the literature and consists in sending a focused pulse to one ion, in the current manuscript one ion at the end of the chain, let the chain evolve so that the excitation transferred to the probe ion travels across the chain, and finally send an opposite pulse to the same ion. The visibility in the Ramsey signal of the internal electronic state of the probe ion can be related, via Fourier-transform, to the normal frequencies of the chain and to their initial thermal state. The measured visibility is compared to the theoretical predictions which include a loss in visibility due to a broad incoherent background of the laser light. The agreement is excellent.

The authors show analytically, although with some approximation, that the visibility is also proportional to the amount of quantum discord, a measure of quantum correlations, between the electronic state, a qubit, and the transverse motion of the probe ion.

The experimental measurement of the visibility in a warm Coulomb crystal is certainly novel, although based on previous proposals. The connection between the observed visibility and discord is novel. However in the latter case, the manuscript lacks some reflection: what are the implications with respect to the quantum correlations of their findings? is there a maximum temperature of the ion chain beyond which entanglement disappears and one has to resort to

discord?

It would be interesting to get an estimate of the initial chain temperature based on the visibility signal $v(t)$. Can the authors comment/show this?

The results are quite interesting for the AMO and Quantum information community, especially those working on quantum correlations. In my opinion, the experimental results achieved in this work open new possibilities that were before limited to theoretical studies. The manuscript is well written and the authors have sufficiently acknowledged the existing literature on the subject.

In summary, I think that this work is an excellent contribution with interesting and novel results. The manuscript lacks some explanations and discussion and I would suggest the authors to improve this. Based on these arguments I strongly recommend it for publication in Nature Communications.

Minor point:

Pag. 4, The sentence: "the pulse duration of 5 - 10 μ s needs to be much shorter than the coupling t_{12} of the local motion" is misleading as the authors are comparing a time with a frequency/energy. Please amend.

Reviewer #3 (Remarks to the Author)

Manuscript reports on precise measurement of phonon autocorrelation function, or quantum discord, in a linear transverse motion of a chain of ions through a precise local coupling to an electronic degree of freedom. The degree of control (also scalability) and accuracy of the measurement seems impressive, as compared to simple theoretical prediction.

I have nevertheless some doubts that the method would work as well also in a genuinely interacting (or non-linear) system? I think demonstrating it in such a regime would be much more interesting, also from fundamental physics point of view. Would any of such extensions be feasible in the present experimental setup?

The paper is definitely worth publishing. But in the present form I cannot recommend it for publication in Nature Communications.

Reviewer #1 (Remarks to the Author):

The authors present a method and experimental results on which they can measure correlation functions of a chain of coupled harmonic oscillators by coupling on of them to a local 2-dimensional quantum system. They work with linear ion chains of variable length trapped in a radio-frequency Paul trap. The radial motion of each ions couples to the others though the coulomb potential. As a local probe the use a the spin degree of freedom of one of the extremal ions which they can control and measure with a strongly focused laser beam.

They can see how an initial excitation propagates along the chain and refocuses on the initial ion at the eigen frequencies of the collective motion.

From a classical perspective the experiment is quite simple minded and easy to understand. Take a long chain of coupled oscillators, kick on of them, extract properties form the whole chain by monitoring only this ion. However the implementation involves the interaction of the electronic state of one of the ions with the local motional modes and generates quantum correlations. As mentioned by the authors, this is an interesting platform in which to study quantum systems coupled to environments, which in the future could be more complex involving defects and coupled environments.

Also, an outstanding feature of this experiment is that they can observe these effects in a “warm” ion crystal where a single quanta of motion is seen propagating along an ion crystal with mean phonon number of 5.

The method seems to be inspired in an extension of 2d spectroscopy techniques applied to ion chains Refs. 21 -23, and a continuation work on what was published by the authors on Ref. 36.

I enjoyed reading the paper. I feel the body of the paper clearly and easily written and tell an interesting coherent story. I found the though, abstract and first two paragraphs, did not flow as easily.

For example, the first sentence. Though the coherence properties of composite quantum systems are of interest to quantum computation, and this paper studies the propagation of a

perturbation in a system of oscillators measured by a 2d quantum system, it seems too much to directly relate the importance of this paper to that. On the other hand, the connection to the study of many-body physics is very well justified.

Following are some specific questions and comments I feel should be addressed.

1) The measurement protocol should be explained somewhere, probably as a subsection in the Methods section. I understand they are using the techniques of ion shelving and fluorescence detection, because of the citation of ref. 33, but some explanation should be available for readers not acquainted with this technique.

- A detailed description of what "excitation" in Fig. 2 is, should be provided, specifically indicating if excitation is measured on the "probe" ion or on the whole chain, or however it was done.

- Also I can't find a proper definition of the quantity P_D , which, to me, is clearly Probability of the ion in the D state after the experiment, but should be explicitly written somewhere.

We agree. We added in the caption of Fig 2: Population of the state

$|D_{1/2}, m_j = -1/2\rangle$ of the probe ion as a function of the detuning.

Further we add to the definition of the visibility in the text below Eq. 3: where $P_D(t)$ is the probability that after the Ramsey sequence with free evolution time t the ion is found in the excited $|D_{1/2}, m_j = -1/2\rangle$ state (see Fig. 1). The visibility $v(t)$ is closely connected to both the autocorrelation function of the probe

In the caption of Fig. 1, we also add: When shining in laser light on the $S_{1/2} \rightarrow P_{1/2}$ transition, observing resulting photons shows that the ion has been projected into the $S_{1/2}$ state whereas recording no photons indicates that the ion has been projected into the $D_{5/2}$ state.

2) The overall quality of the figures and the data presented is a bit messy.

Figure 1 looks like it was prepared in 5 minutes. The font sizes are not balanced and the non clear division into panels makes it a bit uneasy to read.

We readjusted the fonts and their sizes and added clear panels including labeling them with letters.

Figure 3 shows results obtained for 4 different ion chain lengths. I ask myself, what happened in subfigure 3c), why weren't measurements taken for the full time scale. This looks very untidy. I would either recommend the authors to re-measure this case or at the worse to cut the other 3 plots so that the figure doesn't look like "preliminary data".

We plot set now the axes of Fig. 4 to scale.

3) A stylistic comment, which I am not sure about, but the editors will tell.

The sentence “In the supplement, we then find for the modulus of autocorrelation function $|C(t)|$..” doesn’t seem to be in the style results form the Supplementary material are referenced.

We replaced the sentence above with “With these assumptions, we find in Supplementary Note II (see Eqs.~19,22 therein) for the modulus of the correlation function” and changed the headings in the supplement.

3b) Continuing on that sentence, but now on the physics.

There is no discussion of the applicability and universality of relations $|C(t)|$ and $D(t)$ in the main text. This is slipped under the carpet with the reference to the Supp. Mat.

I also wonder, specifically how good are the approximations made to obtain these simple expressions. In particular, the assumption of small excitations. Its validity for a $\pi/2$ pulse, which makes the “expansion parameter” of a value 0.3, as mentioned in the supplementary material is not clear to me. I would like to know, for example if this assumption over or under estimates the results.

Also, I wonder if these assumptions are made for the theoretical predictions plotted with the experimental results, or if these are the full theoretical results. And in such a case, what the deviation is.

We agree. While the required approximations are carefully discussed in the supplementary information, the key ingredients should be summarized more clearly in the main text. We have therefore amended the following sentence:

With these assumptions, we find in Supplementary Note II (see Eqs.~19,22 therein) for the modulus of the

correlation function $|C(t)|$

$$\begin{aligned} \label{corr-func} \end{aligned}$$

$$|C(t)| = |\langle a_1(t) a_1^\dagger(0) \rangle| = (\bar{n} + 1) v(t),$$

$$\end{aligned}$$

where $\bar{n} = \langle a_i^\dagger a_i \rangle \approx 5$ is the average number of phonons in each of the normal modes.

Further, we arrive at a simple measure for the quantum discord (Eq.~32 of Supplementary Note III):

$$\begin{aligned} \label{discord} \end{aligned}$$

$$D(t) \approx \frac{\pi^4}{4} \frac{|C(t)|}{\bar{n} + 1} = \frac{\pi^4}{4} v(t).$$

$$\end{aligned}$$

To derive this relation we have performed a perturbation expansion for small $\theta/2 \approx 0.3$, including contributions of first and second order, and used the additional assumption that the initial phonon state may be approximated by an equilibrium

state of the local phonon modes (for details see Supplementary Note III).

The statements are valid within a perturbative treatment of θ up to second order. However, in general, deviations due to higher-order corrections neither systematically underestimate nor overestimate the result.

The theoretical predictions in Figs. 3 and 4 employ the approximation discussed above. Without such an approximation, the coupled spin-phonon system of up to 42 ions could not be treated theoretically. To add further clarity, we added in the caption of Fig. 3: “The theoretical prediction for the visibility is based on a perturbative expansion (Supplementary Note I)”.

4) In the main text it is not clear what expectation value of n is. The answer can be digged from the supplementary material, but a small hint could be provided.

We added after Eq. 4: where $\bar{n} = \langle a_i^\dagger a_i \rangle \approx 5$ is the average number of phonons in each of the normal modes.

5) The authors state that the initial fast reduction of phase contrast is attributed to a broad incoherent background of the laser light (see methods). The explanation provided seems sound. I wonder though, why for most cases the initial visibility is 0.6 and for the 14-ion chain it is significantly higher.

Yes, this is indeed a justified question. We discovered the reasons for the reduced contrast after finishing the experiments. The parameters for the laser need to be tweaked multiple times per week, sometimes even daily. We discovered that the incoherent background stems from the gain of the feed-back loop. As such the background is very sensitive to the specific gain settings. We observed that it can easily vary by 6 dB or more for reasonable settings of the feed-back parameters.

6) Is no heating considered? Is so, please state why under the experimental conditions this is reasonable.

Motional heating almost exclusively stems from voltage noise at the trap electrodes leading to common noise to all ions. Thus, the center-of-mass mode heats with about 0.2 quanta / ms while all other modes heat with less than 0.01 quanta/ms. Thus, this heating will not influence the intrinsic dynamics of the ion string. One may argue that it should destroy the motion-qubit coherence and thus the visibility, however, much of the motional quantum information is encoded in these other modes (roughly $1/\langle \text{number of ions} \rangle$). Thus, we do not expect to see a significant effect of this heating. These expectations are verified by the good agreement between the theory and the experiments even at large free-evolution times. Note, that the

decoherence of the electronic qubit is on the order of 2 ms (Gaussian model) and thus impacts the coherence stronger.

We added a discussion of this point in the method section:

Another concern is motional heating during the free evolution time. In our trap, motional heating almost exclusively stems from voltage noise at the trap electrodes resulting in common noise to all ions. Thus, the center-of-mass mode heats with about 0.2 quanta / ms while all other modes heat with less than 0.01 quanta/ms. Because the heating is predominantly common mode, motional heating will not influence the intrinsic dynamics of the ion string.

Reviewer #2 (Remarks to the Author):

The manuscript by Abdelrahman & co-workers describes the experimental probing of the phonon dynamics in a one-dimensional ion chain. The probing technique is based on a Ramsey scheme developed a few years ago in the literature and consists in sending a focused pulse to one ion, in the current manuscript one ion at the end of the chain, let the chain evolve so that the excitation transferred to the probe ion travels across the chain, and finally send an opposite pulse to the same ion. The visibility in the Ramsey signal of the internal electronic state of the probe ion can be related, via Fourier-transform, to the normal frequencies of the chain and to their initial thermal state. The measured visibility is compared to the theoretical predictions which include a loss in visibility due to a broad incoherent background of the laser light. The agreement is excellent.

The authors show analytically, although with some approximation, that the visibility is also proportional to the amount of quantum discord, a measure of quantum correlations, between the electronic state, a qubit, and the transverse motion of the probe ion.

The experimental measurement of the visibility in a warm Coulomb crystal is certainly novel, although based on previous proposals. The connection between the observed visibility and discord is novel. However in the latter case, the manuscript lacks some reflection: what are the implications with respect to the quantum correlations of their findings? is there a maximum temperature of the ion chain beyond which entanglement disappears and one has to resort to discord?

It would be interesting to get an estimate of the initial chain temperature based on the visibility signal $v(t)$. Can the authors comment/show this?

We thank the Referee for the comments on our manuscript and the interesting questions raised.

The measurement scheme that is developed and applied in this work allows us to extract information about the spin-phonon discord in a nonequilibrium many-body system with a large

number of excitations in an unbounded Hilbert space. To our knowledge, a method for the extraction of similar information about entanglement is presently not available, which makes it difficult for us to provide viable conclusions on the presence of entanglement in our system. The fact that entanglement is generically more fragile than discord (the entangled states being a subclass of discordant states), we would expect it to be much harder to identify entanglement in such thermally excited systems. Our methods, however, are only able to detect entanglement when operating at zero temperature (where discord and entanglement coincide), and at higher temperatures we indeed have to resort to discord.

Within the validity of the approximations of our theoretical description, the obtained results are in fact independent of the temperature. This means that the tagging mechanism allows us to follow the phononic excitation even on top of a thermal background of a large number of excitations.

While the spin-phonon coupling used to learn about the dynamics of the phononic excitations is strongly affected by temperature, this effect cancels out for determining the visibility as it is measured as a ratio between two instances. Thus, one expects to find only temperature effects if the dynamics of the phononic bath itself is temperature dependent.

We modified the conclusion to clarify these thoughts:

.... Hence, the method allows to follow the dynamics of single phonons in a finite temperature environment. When the spin is coupled to the phonons, its population is strongly affected by temperature. However, within our perturbative treatment, this effect cancels out when determining the visibility as it is measured as a ratio of populations. Thus, we expect to find temperature effects only if the dynamics of the phononic bath itself is temperature dependent.

Our measurements also show that quantum coherence of motional excitations can be preserved even in long ion strings. More precisely, they allow us to extract information about the spin-phonon discord in a nonequilibrium many-body system with a large number of excitations in an unbounded Hilbert space. We note that it appears difficult to extract information about the actual entanglement between the spin and the phonon bath without performing direct measurements on the phonon bath.

Extensions of this work are to measure how phonons

The results are quite interesting for the AMO and Quantum information community, especially those working on quantum correlations. In my opinion, the experimental results achieved in this work open new possibilities that were before limited to theoretical studies. The manuscript is

well written and the authors have sufficiently acknowledged the existing literature on the subject.

In summary, I think that this work is an excellent contribution with interesting and novel results. The manuscript lacks some explanations and discussion and I would suggest the authors to improve this. Based on these arguments I strongly recommend it for publication in Nature Communications.

Minor point:

Pag. 4, The sentence: "the pulse duration of 5 - 10 us needs to be much shorter than the coupling t_{12} of the local motion" is misleading as the authors are comparing a time with a frequency/energy. Please amend.

We added "inverse" in front of coupling t_{12} .

Reviewer #3 (Remarks to the Author):

Manuscript reports on precise measurement of phonon autocorrelation function, or quantum discord, in a linear transverse motion of a chain of ions through a precise local coupling to an electronic degree of freedom. The degree of control (also scalability) and accuracy of the measurement seems impressive, as compared to simple theoretical prediction.

I have nevertheless some doubts that the method would work as well also in a genuinely interacting (or non-linear) system? I think demonstrating it in such a regime would be much more interesting, also from fundamental physics point of view. Would any of such extensions be feasible in the present experimental setup?

The paper is definitely worth publishing. But in the present form I cannot recommend it for publication in Nature Communications.

At the heart of the method lies a phase imprint which connects the created excitation coherently to the probe spin at all times. At the end of the sequence, a measurement which takes only this coherent contribution into account is performed, thereby identifying the original excitation. This technique is indeed reminiscent of and formally equivalent to measurement techniques such as phase cycling or phase matching which are elementary for advanced nonlinear spectroscopic tools. The impressive record of multidimensional spectroscopy from NMR experiments to molecular aggregates up to attosecond nuclear processes indicates that these techniques are able to cope also with highly complex systems in challenging environments. We consider the method presented here as a fundamental step towards the much needed development of a family of advanced spectroscopic techniques for increasingly complex quantum many-body systems. As indicated in the conclusions, future experiments indeed will be able to apply this method to the truly complex setting of a many-body phonon system embedded into an incommensurate or even disordered potential landscape, generated by aptly designed laser fields. We believe that it is crucial to verify and benchmark such a method first under relatively

controlled conditions before venturing into situations where it will be hard to disentangle possible shortcomings of the measurement method from the phenomenology of the dynamics of complex quantum many-body system under study. We therefore believe that our measurements are the first logical and actually meaningful step.

Reviewers' Comments:

Reviewer #1:

Remarks to the Author:

The authors have acted on the observations placed by all referees making the manuscript much better.

I feel it is now sound and in shape for publication.

The comments of referee 3, and re reading the manuscript reminded me of a recently published paper, which I'd like to bring the authors to notice. I feel they will find it interesting as it shares some ideas with their manuscript and could be inspirational for future applications of the presented scheme.

Noise-induced transport in the motion of trapped ions
Cecilia Cormick and Christian T. Schmiegelow
Phys. Rev. A 94, 053406 – Published 9 November 2016

Reviewer #2:

Remarks to the Author:

In the revised version the authors have successfully addressed all the points raised in my previous report. I find that the quality of the manuscript has improved considerably and therefore I recommend the manuscript for publication in Nature Communications without further changes.

Reviewer #3:

Remarks to the Author:

After very helpful reply from the authors, together with the comments and answers of the other referees, I conclude that the manuscript indeed represents a considerable step forward in terms of an experimental method, even though from a theoretical viewpoint and the nature of the model studied, it seems quite rudimentary.

I have thus no objections against publication of this work in Nature Communications.